# Development of a dynamic prediction model for unplanned ICU admission and mortality in hospitalized patients

**Davide Placido**[1], **Hans-Christian Thorsen-Meyer**[1,2], **Benjamin Skov Kaas-Hansen**[2,3], **Roc Reguant**[1,4], **Søren Brunak**[1,5]*

**1** Novo Nordisk Foundation Center for Protein Research, University of Copenhagen, Denmark, **2** Department of Intensive Care Medicine, Copenhagen University Hospital, Rigshospitalet, Copenhagen, Denmark, **3** Section for Biostatistics, Department of Public Health, University of Copenhagen, Denmark, **4** Australian e-Health Research Centre, Commonwealth Scientific and Industrial Research Organisation, New South Wales, Sydney, Australia, **5** Copenhagen University Hospital, Rigshospitalet, Copenhagen, Denmark

* soren.brunak@cpr.ku.dk

**Data Availability Statement:** The software is available online at https://github.com/daplaci/ClinicalDeteriorationNet. The authors do not have

## Abstract

Frequent assessment of the severity of illness for hospitalized patients is essential in clinical settings to prevent outcomes such as in-hospital mortality and unplanned admission to the intensive care unit (ICU). Classical severity scores have been developed typically using relatively few patient features. Recently, deep learning-based models demonstrated better individualized risk assessments compared to classic risk scores, thanks to the use of aggregated and more heterogeneous data sources for dynamic risk prediction. We investigated to what extent deep learning methods can capture patterns of longitudinal change in health status using time-stamped data from electronic health records. We developed a deep learning model based on embedded text from multiple data sources and recurrent neural networks to predict the risk of the composite outcome of unplanned ICU transfer and in-hospital death. The risk was assessed at regular intervals during the admission for different prediction windows. Input data included medical history, biochemical measurements, and clinical notes from a total of 852,620 patients admitted to non-intensive care units in 12 hospitals in Denmark's Capital Region and Region Zealand during 2011–2016 (with a total of 2,241,849 admissions). We subsequently explained the model using the Shapley algorithm, which provides the contribution of each feature to the model outcome. The best model used all data modalities with an assessment rate of 6 hours, a prediction window of 14 days and an area under the receiver operating characteristic curve of 0.898. The discrimination and calibration obtained with this model make it a viable clinical support tool to detect patients at higher risk of clinical deterioration, providing clinicians insights into both actionable and non-actionable patient features.

permission to share the data directly; following ethical approval data can be made available for use in secure, dedicated environments via application to the Danish Regions and the Danish Health Data Authority. Researchers wanting access to the data and to use them for research will be required to meet research credentialing requirements. This study was approved by the Danish Patient Safety Authority (3-3013-1731 and 3–3013–1723), the Danish Data Protection Agency (DT SUND 2016–48, 2016–50, 2017–57 and UCPH 514-0255/18-3000:) and the Danish Health Data Authority (FSEID 00003092, FSEID 00003724, FSEID 00004758 and FSEID 00005191).

**Funding:** We would like to acknowledge the Novo Nordisk Foundation (grants NNF17OC0027594 and NNF14CC0001) and the Danish Innovation Fund (5153-00002B) which supported this study. These foundations contributed to the financing of salaries for SB, DP, RR. BSK-H acknowledges funding from "Grosserer Jakob Ehrenreich og Hustru Grete Ehrenreichs Fond". The funders had no role in study design, data collection and analysis, decision to publish, or preparation of the manuscript.

**Competing interests:** I have read the journal's policy and the authors of this manuscript have the following competing interests: SB has ownership in Intomics A/S, Hoba Therapeutics Aps, Novo Nordisk A/S, Lundbeck A/S, ALK Abello and managing board memberships in Proscion A/S and Intomics A/S.

## Author summary

Early warning scores are used in the hospital to assess the severity of patient's conditions, with higher score values indicating patients at higher risk of clinical deterioration. These scores are traditionally based on a relatively small set of variables, and they require medical personnel to collect the measurements included in the score calculation. In this study we investigated whether machine learning models using data routinely registered in electronic health record systems of hospitals can predict clinical deterioration. The outcome was defined as in-hospital mortality and unplanned transfer to the intensive care unit. Data included disease history, laboratory data, and medical notes from 12 hospitals of the Capital Region of Denmark and Region Zealand during the period 2011–2016. The model had good discrimination and calibration, with better performance when assessing the risk every 6 hours during an admission and using a prediction window of 14 days. The model was able to identify abnormal lab values and in general variables associated with critical conditions. Our study shows that machine learning models like the one developed here serve as clinical decision support tools for this purpose and are ready to be prospectively validated in the clinical setting.

## Introduction

Evidence-based medicine is at the foundation of the clinical decision process: doctors must continuously take decisions using their knowledge to provide the best care to the patients. For this reason, numerous scores are used in healthcare to support clinicians with decision-making. Such scores vary depending on the target population and the intended use, therefore each medical specialty has its own examples (e.g. in the ICU, APACHE [1] and SAPS [2] scores are commonly used to assess severity of illness).

In general departments, Early Warning Scores (EWSs) are used to assess the health status of hospitalized patients. The first EWS was based solely on five physiological parameters, and was later updated by adaptations and improvements [3,4]. VitalPAC was one such early warning score (ViEWS), for which modifications were introduced [5]. These modifications were based on clinicians' knowledge about the relationship between physiological data and adverse clinical outcomes. Further modifications were implemented in the national early warning score, NEWS, currently being the most used in the Capital Region of Denmark [6].

These scores are used extensively because their simplicity makes them applicable and easily comparable across different departments and countries. They specifically provide estimates for the risk of adverse outcomes [7] such as cardiac arrest, ICU transfer, and in-hospital mortality. However, they suffer from some limitations. The relatively few data types used limit their predictive power. Relevant features like biochemical measurements, the order of temporality in the data, or non-linear feature interactions are ignored. These risk calculations are made without exploiting all the information that current EHR systems provide [8]. Moreover, the scores mostly rely on data that are manually collected by the staff. This often entails a higher number of missing and incorrect data items, as compared to a fully automated approach. Recently it was shown in population-wide data from inpatients of the Capital Region of Denmark, that around 10% of the NEWS records were incomplete and 0.2% had implausible values [9].

To improve prediction accuracy and reduce alarm fatigue, new applications using more data and more sophisticated methodologies have been developed [10]. Given the heterogeneous nature of EHR data and the rarity of events such as clinical deterioration, machine learning methods seem well-suited for addressing this task. Deep learning, in particular, has

performed well for many similar clinical tasks. For example, Shamout et al. [11] developed a deep learning model that uses vital signs to predict the composite outcome of ICU transfer, cardiac arrest and mortality. Cho et al. [12] developed a deep learning model to predict the composite outcome of cardiac arrest and ICU transfer, while da Silva et al. [13] developed a long-short term memory neural network (LSTM) to predict the worsening of vital signs. A recurrent deep neural network has also been used by Tóth et al. [14] to predict stability of vital signs to avoid unnecessary measurements at nighttime. Similarly, dynamic deep learning models using multiple data sources have been developed for different tasks; Thorsen-Meyer et al. [15] constructed one to predict 90-days mortality after ICU admission, Rajkomar et al. [16] to predict in-hospital death, 30-day unplanned readmission, prolonged length of stay and patient's final discharge diagnoses; Tomašev et al. [17] to predict kidney failure; and Lauritsen et al. to predict risk of sepsis and acute critical illness [18,19].

Although deviating vital signs often precipitate clinical deterioration, their recognition in the general ward requires continuous engagement by health care personnel, which can delay or even preclude their recording. Often, vital signs are not included systematically in patient records in structured form; this was also the case in the data foundation for this study. Therefore, the hypothesis in this study is that a deep learning model using heterogeneous clinical data collected less frequently is able to predict at regular intervals during an admission the risk of the composite outcome of in-hospital death and unplanned ICU transfer (Fig 1). An important aspect is at the same time to inform the clinicians of features which are actionable.

To this end, we combined natural language processing algorithms and recurrent neural networks (Fig 2) to leverage latent patterns in the data. We subsequently assessed the impact of

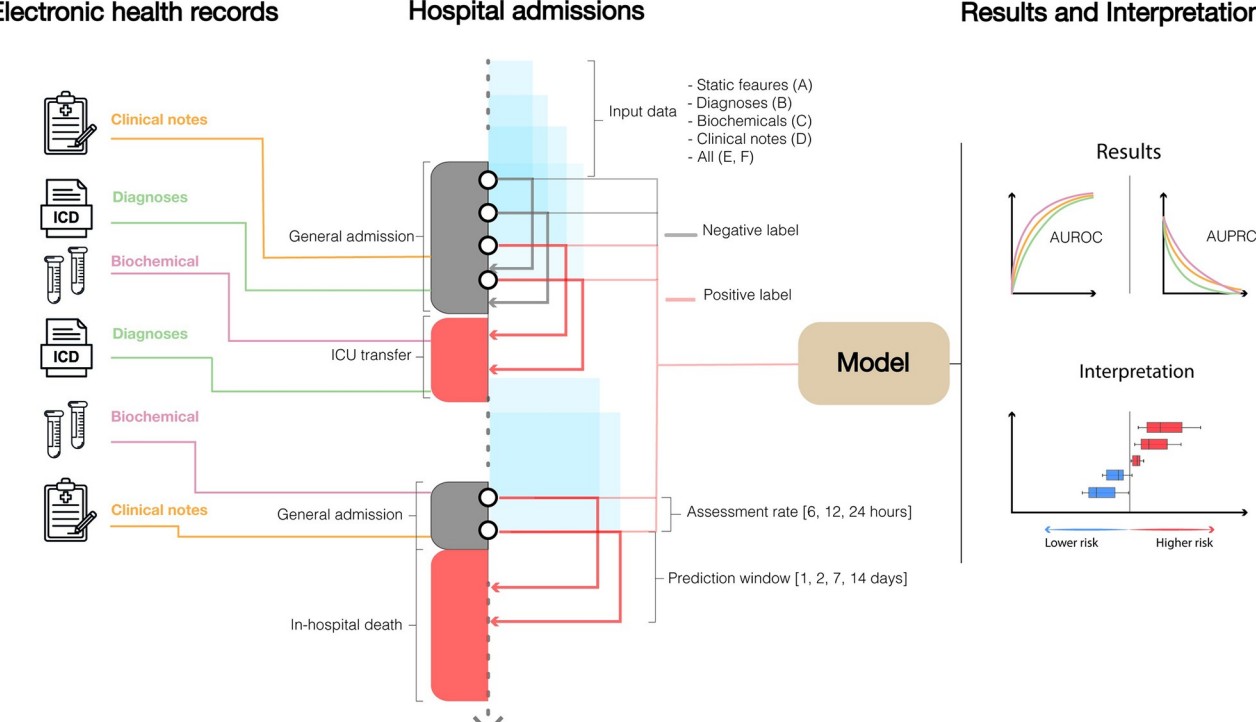

**Fig 1. General structure of the prediction framework.** Given a specific assessment rate (time between two consecutive risk assessments) and prediction window (time window within which the outcome is observed), the risk of clinical deterioration was assessed continuously during each general admission. All the data up to the time of assessment was used to train the model, which was then evaluated and interpreted on the holdout test data.

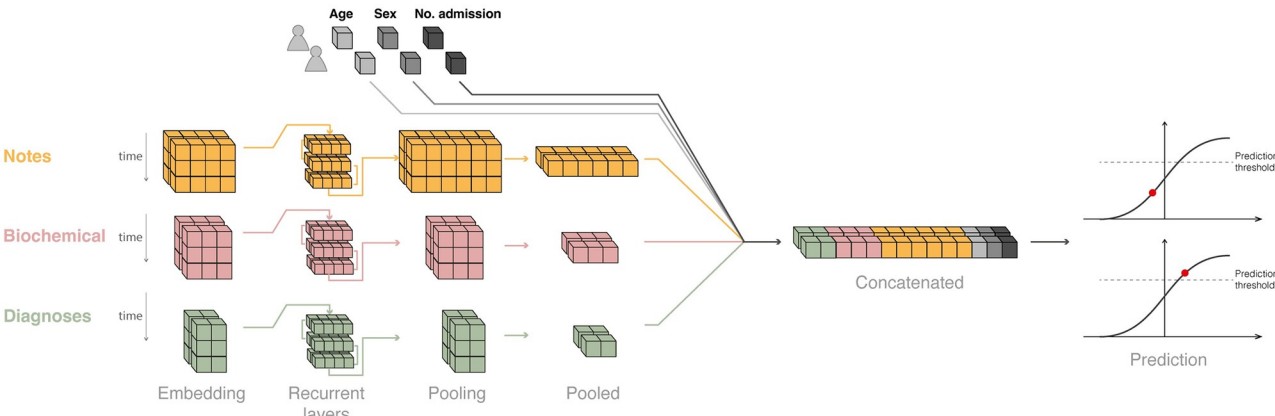

**Fig 2. Structure of the deep learning model, exemplified by two admissions.** For each sample (i.e. admission at a specific time point) a 2D tensor (matrix) comprises the sequence of embedded tokens until the time of assessment. Each tensor was given as input to a recurrent block (GRU or LSTM, part of the hyperparameters search) and the time dimension was pooled through an attention pooling layer. The flattened tensors were then concatenated and fed to a linear layer with a standard logistic activation function.

the single data sources and the so-called tokens extracted from each of them to make the model explainable across features. We successfully validated the model internally on a patient subset selected randomly from the pool of patients available.

## Methods

This paper adheres to relevant items in the Transparent Reporting of a multivariable prediction model for Individual Prognosis or Diagnosis statement (TRIPOD) (S4 Table) [20].

### Patients and outcome

The data comprise all inpatient admissions, from 2011 to 2016, to 12 public hospitals in the Capital Region of Denmark and Region Zealand. The admissions were pieced together by concatenating consecutive inpatient visits 24 hours apart so that department transfers were not considered two separate events.

ICU admissions not preceded by any other admission within the last 24 hours, outpatients and acute admissions were excluded, as were individuals with disconnected medical record history (either because these patients moved to another country or lacked a stable residence) and minors (age <16 years) (Fig 3).

The outcome *clinical deterioration* was defined as unplanned ICU transfer and in-hospital mortality within the so-called prediction window. Unplanned ICU transfer was defined as any acute admission to an ICU within 24 hours after discharge from a non-ICU ward.

### Model development

The model architecture (Fig 2) was designed as a scalable network that can be adapted to different data modalities by using one sub-model per data domain; each sub-model consists of an embedding layer (which transforms the categorical variables into vectors), a recurrent neural network (which learns from the sequence of embedding vectors) and a pooling layer (which reduces the vectors' dimensionality).

The network uses tokens, i.e. sequences of characters grouped together based on their semantics, as input. We chose the entity embedding approach to exploit the heterogeneity and sparseness of the input data, thus mapping the tokens constructed from the categorical features

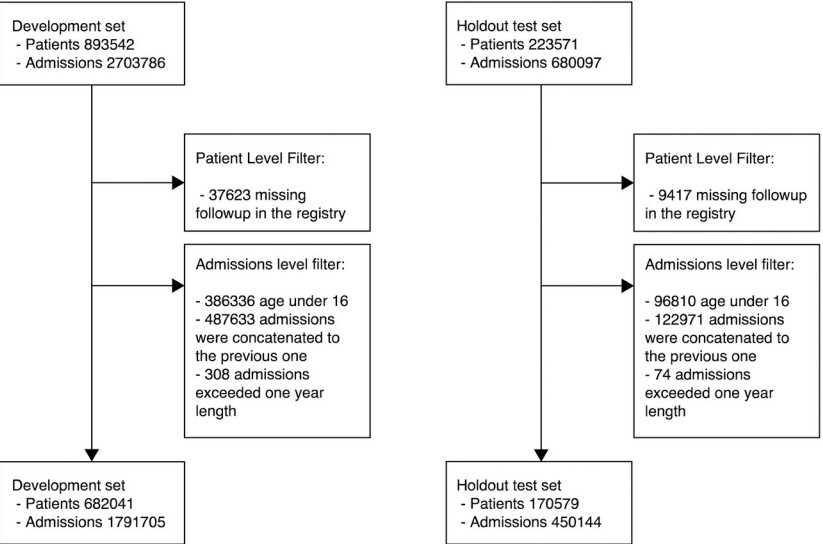

**Fig 3. Attrition diagram: Study profile.**

to embeddings in a Euclidean space (i.e. the embedding space) [21]. For each sub-model, the vocabulary size V (i.e. the number of unique tokens for the specific data source) was directly used to calculate the size of the embedding representation according to $EmbeddingSize = 6 \cdot \alpha \cdot \sqrt[4]{V}$, where the embedding coefficient α was estimated by hyperparameter search [16].

We explored two types of recurrent neural networks, Gated Recurrent Unit (GRU) [22] and Long Short-Term Memory (LSTM) networks [23]. Rather than max or average pooling—which retains neither positional nor intensity information—we used an attention-based pooling [24] which employs a weighted mechanism to retain the most relevant parts of a sequence. The attention-based pooling layer was used to aggregate the temporal dimension of the recurrent layer output, which was then concatenated across the different data sources. The final layer was a linear (= dense) layer with a standard logistic activation function hence mapping the output into the interval [0, 1] to yield valid predicted probabilities. Features without temporal components (age, number of previous hospital admissions and sex) were fed into the model by concatenating them to the output of the pooling layer.

## Data sources and processing

The input data comprised medical history, biochemical measurements, clinical notes and demographics (age, sex and number of previous admissions). A schematic of the time spans for the different data modalities is depicted in S1 Fig.

The medical history was extracted from the Danish National Patient Registry (DNPR) for the period between 1977 and 2018 [25]. DNPR is a nation-wide registry which covers essentially all hospital encounters in Denmark. The disease codes use Danish adaptations from SKS of the International Classification of Diseases (ICD) version 8 up to 1993, and ICD-10 from 1994 and onwards. Only diagnosis codes recorded prior to the admission date were included in the predictive schemes. No manual curation was carried out on the disease history data as quality checks are handled internally by the DNPR (positive predicted values for registered ICD codes varies according to time and clinical specialty [25,26]). Summary statistics on

comorbidities (Table 1) were calculated from all chapters excluding XIX and XXI (injuries and administrative codes), while all codes were included in model training.

Biochemical values were extracted from the Clinical Laboratory Information System (LABKA) and the Clinical Chemistry Laboratory System (BCC) databases [27] for the period overlapping the hospital admissions (2011–2016). These databases collect all the biochemical tests performed by the Danish hospital laboratories in the Capital Region and Region Zealand, respectively. Before constructing the tokens, the data sets were manually curated as described in Muse et al., [28]: lookup dictionaries were used to standardize components, specimens' names and units; dictionaries were also used to identify incomplete, unreadable or failed tests. Moreover, they were manually curated to identify tests having quantitative and qualitative results and to correct them for typos or wrong numerical notation. The biochemistry tokens were constructed using the name of the biochemical component, the specimen (blood, plasma etc.), the unit and the quantile of the value of the measurement (to yield a vocabulary with a reasonable size); for example, *HEMOGLOBIN_B_mmol/L@8-8.5*. The quantile binning was included as a hyperparameter.

**Table 1. Summary statistics for the development and test sets.** Statistics are calculated at the admission level. Counts are given as well as medians when numerical distributions are provided, shown in the format Median [25th-75th quantile].

|  | Development Set | Test Set |
|---|---|---|
| Number of Patients | 682041 | 170579 |
| Number of Admissions | 1791705 | 450144 |
| Age at prediction |  |  |
| Discharge | Median 60 [40–73] | Median 60 [40–73] |
| ICU Transfer | Median 67 [56–75] | Median 67 [56–75] |
| In-hospital Death | Median 78 [68–86] | Median 78 [69–86] |
| Sex: |  |  |
| Male | 798185 (45%) | 199732 (44%) |
| Female | 993520 (55%) | 250412 (56%) |
| Outcome |  |  |
| Discharge | 1752329 (97.8%) | 440221 (97.8%) |
| ICU Transfer | 13129 (0.7%) | 3305 (0.7%) |
| In-hospital Death | 26247 (1.5%) | 6618 (1.5%) |
| Number of previous admissions* |  |  |
| Discharge | Median 1 [0–4] | Median 1 [0–4] |
| ICU Transfer | Median 2 [0–6] | Median 2 [0–6] |
| In-hospital Death | Median 3 [1–7] | Median 3 [1–6] |
| Length of stay before outcome (hours)*: |  |  |
| Discharge | Median 31 [9–98] | Median 31 [9–97] |
| ICU Transfer | Median 35 [8–121] | Median 35 [8–132] |
| In-hospital Death | Median 126 [43–285] | Median 122 [43–279] |
| Type ICU Transfer |  |  |
| Surgical | 2358 (18%) | 639 (19.3%) |
| Medical | 10771 (82%) | 2666 (80.7%) |
| Number of previous diagnoses | Median 23 [14–36] | Median 23 [14–36] |
| Number of tokens from laboratory available: |  |  |
| @ 24 hours | Median 19 [0–36] | Median 19 [0–35] |
| @ 48 hours | Median 28 [8–66] | Median 29 [8–66] |
| @ 72 hours | Median 38 [16–66] | Median 38 [16 66] |
| Number of tokens from clinical notes available: |  |  |

*(Continued)*

**Table 1.** (Continued)

| | Development Set | Test Set |
|---|---|---|
| @ 24 hours | Median 215 [113–344] | Median 215 [112–345] |
| @ 48 hours | Median 316 [190–480] | Median 316 [189–481] |
| @ 72 hours | Median 396 [246–593] | Median 396 [245–594] |
| Most common comorbidities (ICD-10 code): | | |
| Abdominal and pelvic pain (R10.0) | 563378 (31.44%) | 143010 (31.77%) |
| Essential hypertension (I10) | 470525 (26.26%) | 117239 (26.04%) |
| Type2 diabetes (E11) | 378979 (21.15%) | 92135 (20.47%) |
| Mental and behavioral disorders due to use of alcohol (F10) | 377978 (21.1%) | 93106 (20.68%) |
| Chronic ischemic heart disease (I25) | 365072 (20.38%) | 91999 (20.44%) |
| Admitting department | | |
| Internal medicine | 253743 (14.2%) | 64173 (14.3%) |
| Emergency medicine | 244510 (13.6%) | 61219 (13.6%) |
| Gynecology and obstetrics | 190047 (10.6%) | 47597 (10.6%) |
| Orthopedic surgery | 139427 (7.8%) | 34761 (7.7%) |
| Cardiology | 138989 (7.8%) | 34593 (7.7%) |
| Surgical gastroenterology | 129420 (7.2%) | 32487 (7.2%) |
| Urology | 88387 (4.9%) | 21769 (4.8%) |
| Miscellaneous | 607182 (33.9%) | 153545 (34.1%) |

* for all the admissions having one of the outcome listed below.

The clinical notes written by physicians were extracted from the EHR data for the admissions from 2011 to 2016. The free text required some preprocessing before tokenization, such as removing punctuation, names, stop words [29], negations and signatures of the clinicians. Due to the large number of terms coming from the medical notes, the embedding of this specific data source was trained separately on the full corpus using fastText [30], to reduce the number of parameters to update during training.

Because we did not force our data into a tabular format, no data were imputed because only observed values were used. As is typical in NLP approaches, sequences were padded to a specific length (the optimal length for each data source was learned during the hyperparameter search). Sequences longer than the defined padding size were truncated; otherwise, they were extended with the padding token 'PAD'. To control for very rare tokens (which may be outliers or very rare biochemical tests) a cut-off ("min_freq") was learned during hyperparameter search; tokens with a frequency below min_freq were replaced with an "UNK" (unknown) token.

## Training and evaluation

We randomly split the dataset into a development set (80%) for model creation and an independent holdout test set (20%) for model internal validation. The split of the dataset was done at the patient level and assigned admissions of the same patient to the same set to avoid leaking information between the sets [31]. The development set was further split into a training set (80%, 64% of total) and validation set (20%, 16% of total) to counter overfitting and to calibrate the model before testing it on the holdout set (S8 Fig).

Three submodels were trained separately to find the best submodel-specific hyperparameters; we used Optuna's multivariate TPEsampler, based on the Three-structured Parzen

Estimator (TPE) algorithm to search the hyperparameter spaces [32,33]. While searching the hyperparameter space, we fixed the prediction window at 24 hours and the assessment rate at 12 hours, to facilitate the comparison across the different data types. S1 Table shows the hyperparameters explored for each submodel; S2, S3 and S4 Figs illustrate the hyperparameter searches. For each experiment, the loss on the validation set was used as training metric to select the model at the best epoch. The performance metric used to select the best model for each search was the area under the precision-recall curve (AUPRC). Although the area under the receiver operating curve (AUROC) score is a more common metric for classification tasks, in this case it would not be sufficient to appreciate the real ability of the model to discriminate between the two classes due to their considerable imbalance [34]. AUPRC on the other hand is much more robust to class imbalances. We used 200 bootstraps samples [35] to construct the 95% confidence interval for the metrics shown in figures and tables.

Using the best submodel architectures, we trained and evaluated an ensemble model for different prediction windows (1, 2, 7, 14 days) and assessment rates (every 6, 12 and 24 hours). Finally, we applied post-hoc isotonic calibration to align the final predicted risk with the actual outcome incidence, using the data in the validation set [36]. The model re-calibration was achieved by fitting an isotonic regression using the output of the model prior to calibration as regressor and the actual label of the samples as response variable. We used the validation set to fit the isotonic regression and kept the test set untouched. The fitted isotonic model was used to adjust the output of the uncalibrated model to get calibrated predictions.

All the results reported are generated using the re-calibrated model on the test set. To control for biases driven by age or sex, performances were also evaluated at each time of assessment for the different subgroups.

## Interpretation

The impact of the different tokens on the model outcome was calculated using the Gradient-Shap algorithm [37] from the *Captum* library [38]. Given an input feature of a single risk assessment, its Shap value is correlated to how much (and in what direction) that feature pulls the individual-level prediction away from the population-level mean risk. Importantly, Shap values do not represent the effect of a single feature on the model outcome but rather the effect of that feature in the context of a coalition of features. Shap values were calculated for the best model after isotonic re-calibration.

## Results

The model was trained on 682,041 unique patients and 1,791,705 admissions and evaluated on 170,579 patients and 450,144 admissions as described in Methods (S3 Table). In both parts, 1.5% of the admissions resulted in in-hospital mortality and 0.7% in ICU transfer. 2,583 tokens from the ICD code data type (medical history), 2,421 tokens from the biochemical measurements and 403,869 tokens from the medical notes were used as input. We investigated the inclusion of each data type separately as well as jointly in the same model.

When we explored how the prediction window and assessment rate affect the performances, the most performant model based on the AUPRC was the one trained on all the data sources using a prediction window of 14 days and an assessment rate of 6 hours with an AUROC of 0.904 [0.903–0.904] and AUPRC 0.285 [0.283–0.286] (Fig 4, Table 2). This model was after isotonic recalibration well-calibrated (Fig 4C, S5 Fig) with a calibration slope of 0.964 [0.951, 0.98], an intercept of 0.002 [0.001–0.003] and an upper bound risk of 79%. A full list of precision, recall and specificity for the different models is in the Supplementary (S2 Table).

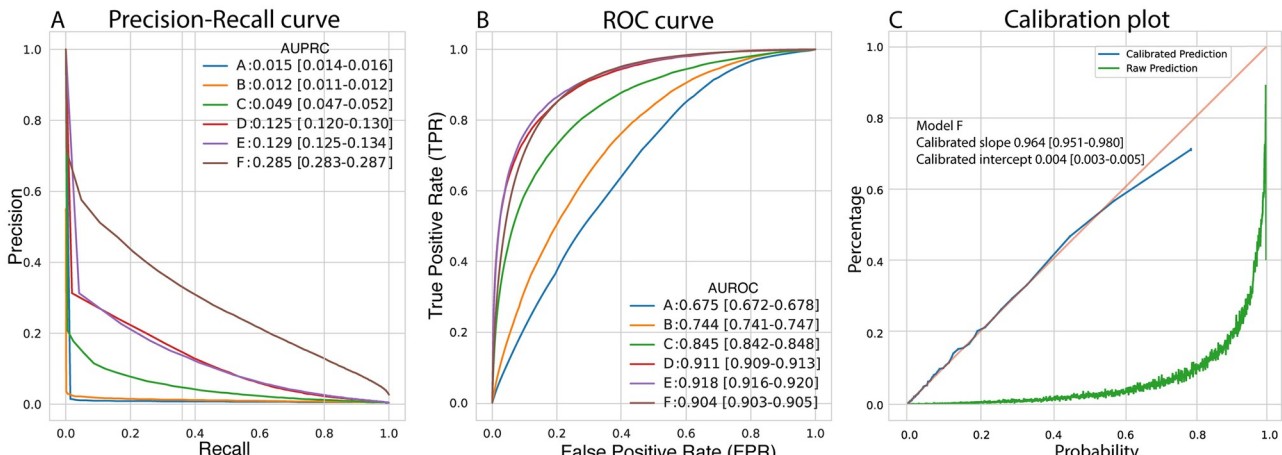

**Fig 4. Performance of the model for prediction of unplanned ICU transfer or in-hospital death.** Panel A: precision recall curves show the values of precision and recall at different thresholds for the six models. Panel B: receiver operating characteristic (ROC) curves show the values of true positive rate (= recall = sensitivity) and false positive rate (= 1-specificity) at different thresholds for the six models. Model A: age, sex, number of admissions. Model B: model A + medical history data. Model C: model A + biochemical data. Model D: model A + clinical notes. Models E and F: model A + medical history data, biochemical data and clinical notes. Models A–E use a 24-hour prediction window and a 12-hour assessment rate. Model F uses a 14-day prediction window and a 6-hour assessment rate. Panel C: calibration curve (observed fraction of positive sample at different predicted probability) for Model F before and after isotonic calibration.

The performances were similar across sexes, with AUPRCs of 0.288 [0.285–0.290] and 0.280 [0.277–0.283] for males and females, respectively. The performances for the different age groups varied more, with a trend of AUPRC increasing with age (age 16–37: 0.168 [0.157–0.180], age 37–58: 0.252 [0.246–0.257], age 58–79: 0.277 [0.274–0.280], age 79–100: 0.310 [0.307–0.313]).

We observed an increased AUPRC for risk estimates further into the hospital stay with a peak at 7 days into the admission (S6 and S7 Figs).

Among the models trained separately using a 24-hour prediction window and 12 hours assessment rate, the most performant was the one trained on the clinical notes with an AUPRC of 0.125 [0.120–0.130]; this value improved to AUPRC of 0.129 [0.125–0.134] when the model was trained on all the data using the same prediction window and assessment rate.

**Table 2. Performance on the test set for models using all data types.**

| Assessment rate | Prediction window | AUROC | AUPRC | Calibration | |
|---|---|---|---|---|---|
| | | | | Slope | Intercept |
| 6h | 1d | 0.925 [0.923–0.926] | 0.152 [0.149–0.155] | 1.042 [1.02–1.064] | -0.001 [-0.001–0.0] |
| | 2d | 0.922 [0.921–0.923] | 0.180 [0.177–0.184] | 0.988 [0.966–1.007] | 0.002 [0.001–0.003] |
| | 7d | 0.913 [0.912–0.914] | 0.269 [0.267–0.271] | 0.996 [0.985–1.008] | 0.002 [0.001–0.003] |
| | 14d | 0.904 [0.903–0.904] | 0.285 [0.283–0.286] | 0.964 [0.951–0.98] | 0.004 [0.003–0.005] |
| 12h | 1d | 0.918 [0.916–0.920] | 0.129 [0.125–0.134] | 0.998 [0.96–1.045] | 0.001 [0.0–0.002] |
| | 2d | 0.916 [0.914–0.917] | 0.187 [0.183–0.191] | 0.996 [0.977–1.021] | 0.002 [0.0–0.003] |
| | 7d | 0.898 [0.897–0.899] | 0.233 [0.231–0.236] | 1.045 [1.026–1.067] | -0.001 [-0.002–0.0] |
| | 14d | 0.901 [0.900–0.902] | 0.278 [0.275–0.282] | 0.952 [0.912–0.997] | 0.005 [0.001–0.007] |
| 24h | 1d | 0.905 [0.901–0.908] | 0.135 [0.130–0.140] | 1.038 [0.992–1.078] | 0.0 [-0.001–0.001] |
| | 2d | 0.901 [0.899–0.903] | 0.163 [0.157–0.169] | 1.017 [0.969–1.059] | 0.0 [-0.001–0.002] |
| | 7d | 0.896 [0.894–0.897] | 0.232 [0.227–0.236] | 1.014 [0.969–1.05] | 0.001 [-0.001–0.003] |
| | 14d | 0.893 [0.892–0.894] | 0.250 [0.246–0.254] | 0.974 [0.951–1.006] | 0.003 [0.001–0.005] |

The best encoding of disease diagnoses was rolling diagnoses up to the third ICD level (e.g. C341M to C34) and a padding size of 9 (S2 Fig). The optimal quantile resolution for the biochemical data was deciles (i.e. 10 bins) and a padding size of 28 was the optimal number of lab values to include before time of prediction (S3 Fig). The optimal padding size for medical notes was 299, reflecting the larger amount of information usually held by the clinical notes (S4 Fig). The complete list of the optimal parameters for each search can be found in the supplementary S1 Table.

### Feature importance

The importance of the tokens from the different vocabularies can be sorted according to their attributed Shap values (Fig 5). Higher age and male sex were associated with elevated risk of deterioration. The opposite was the case for the number of previous admissions: a higher number of previous admissions was associated with lower risk of deterioration (Fig 5A–5C).

The diagnosis tokens associated with clinical deterioration were often those of acute illnesses such as respiratory failure, neoplasm and pneumonia (Fig 5D). In contrast, diagnosis codes related to pregnancy, chronic conditions and orthopedics were associated with lower risk.

Low levels of albumin, lymphocytes and sodium were associated with elevated risk of clinical deterioration; the same were high levels of C-reactive protein, leukocytes, sodium, lactate dehydrogenase, potassium and carbamide (Fig 5E). In contrast, normal levels of leucocytes, hemoglobin, C-reactive protein, sodium and potassium were all associated with low risk of deterioration.

The tokens from clinical notes most strongly associated with clinical deterioration were *severely*, *intensive*, *respiratory and chronic*. *Pain*, *leukocytes*, *control*, *normal*, *home*, *discharge(d)* on the other hand were all associated with low risk (Fig 5F).

## Discussion

The main aim of this study was to explore whether data types registered routinely in general departments are predictive of clinical deterioration, ultimately to assess if these suffice for this task. A solution based on data collected routinely might circumvent a common weakness of EWS, i.e. they depend on data that require clinical engagement (e.g., vital signs). Consequently, offering a viable alternative for risk stratification with minimum additional manual data collection effort is preferable.

Exploiting the combined power of entity embedding of tokens from electronic health records and the ability of recurrent neural networks to learn temporal patterns from such data, we built a performant (AUROC and AUPRC up to 0.90 and 0.29, respectively) and well-calibrated deep learning model for predicting the risk of clinical deterioration. Specifically, leveraging medical history data (up to 40 years) from a national register along with in-hospital biochemical data and clinical notes, we trained the model dynamically, meaning that the same model can handle different time points for the same admission.

Direct comparison of our model to the performances of NEWS or MEWS is not possible, since the vital signs used for the calculation of such scores are not collected in our EHR dataset. Comparison of the performance of our model to other work is also non-trivial: AUROC is the most common metric for risk stratification and usually the one used to compare performance across studies. Nevertheless, it is unsuited for imbalanced prediction problems (such as the one defined here) because it disregards the prevalence of the outcome of interest [34]. Although AUPRC does account for prevalence, it is not always reported in the classic studies on EWS scores. Moreover, direct comparison of the AUPRCs requires equal (or at least similar) prevalence of the outcome across studies; this is problematic because the prevalence tends to vary between cohorts, and, more importantly, so do the criteria used to define the outcome.

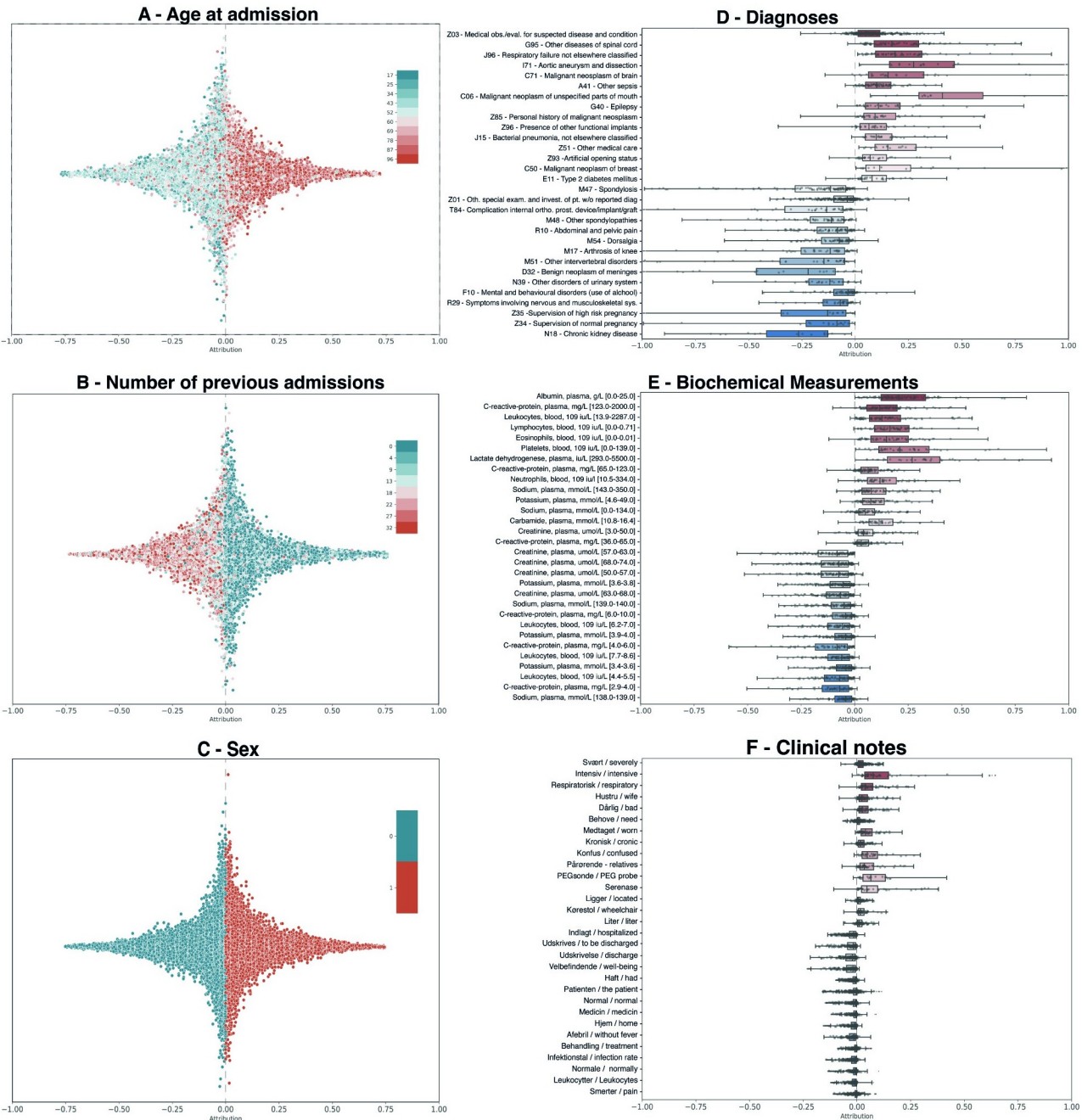

**Fig 5. Contribution of the different tokens to the outcome of clinical deterioration.** Shap values were estimated using the best model (model using age, sex, number of admissions, medical history data, biochemical data and clinical notes with a 14-day prediction window and 6-hours assessment rate). Panels A-C: the distribution of Shap values for each feature without temporal components, with the color scale representing the feature value. Panels D-F: the distribution of Shap values for top-15 tokens (in up- and downward directions, respectively). Overlain boxes-and-whiskers show medians, quartiles and 1.5 x quartiles.

For example, Watkinson et al. [39] defined the outcome as the composite of ICU transfer, in-hospital death and cardiac arrest with a prediction window of 24 hours (AUROC of 0.868 [0.864–0.872]). Dziadzko et al. [40] defined the outcome as in-hospital death or respiratory failure, with a prediction window of 48 hours (AUROC 0.87 [0.85–0.88] and 0.90 [0.84–0.95]

in 2013 and 2017, respectively). Malycha et al. [41] defined the outcome as in-hospital death and ICU transfer within 24 h following assessment from patients admitted longer than 24 hours (AUROC 0.823 [0.819–0.824]). Cho et al. [12] instead defined the outcome as cardiac arrest and unexpected ICU admission, occurring within 0.5–25 hours from the assessment (AUROC 0.865 [CI not reported]).

The performance difference of models A-E (Fig 4) is driven by information learnt from different data types. The performance does not increase linearly with the addition of new data types, suggesting substantial overlap in the latent information of diagnosis codes, biochemical data and clinical notes. This corresponds well to what one would expect, e.g. a clinical note may very well cite lab values (and likely those of greatest clinical interest) and summarize anamnestic information such as comorbidities also registered in the diagnosis code data.

On the other hand, the change in performance from model E to model F (AUPRC from 0.129 [0.125–0.134] to 0.285 [0.283–0.287]) is likely driven by the different incidence of the outcome when using longer prediction windows, essentially making the prediction task easier because there are more examples to learn from. Also, it is much more difficult to discriminate between patients having a severe outcome in the subsequent 24 or 48 hours, because drivers of short-term mortality/ICU transfer depend not only on the physiological status of the patient but also on many factors beyond what is captured in clinical data, e.g. the coordination of the resources within and between hospital departments. The assessment rates employed are consistent with how often new observations and information are recorded in patient files: new clinical notes and biochemical measurements are usually recorded a couple of times every day and normally at least once. The change in AUPRC and AUROC during the admissions reflects the increasing availability of new records when the patients proceed further into the hospitalization (S6 and S7 Figs), and that patients tend to deteriorate either early or late in the admission and less in between.

We decided to keep the outcome definition broad rather than predicting specific severe conditions such as sepsis or organ failure, notoriously difficult to operationalize for prediction tasks [42]. In contrast, using a general definition of clinical deterioration allowed us to keep the feature space more generic and less dependent on specific illnesses or patients subgroups. Indeed, our model seeks not to advice on interventions but to flag patients at risk of (more of less imminent) clinical deterioration so health care staff can intervene in a manner appropriate for the patient in question, hopefully translating into improved prognosis for that patient.

An ensemble structure of the network was preferred over a structure in which the different data types contributed to the creation of the same embedding space. Separate submodels (embedding + recurrent linear layer + pooling) allowed us to tune their architectures; optimal padding sizes and embedding coefficients, for example, differ for diagnosis codes and biochemical data. Its ensemble nature also renders the model scalable for incorporating new data types/domains.

To obtain a reasonably well-defined training cohort applicable to as many patients as possible, we included all in-patient admissions to any hospital department, excluding only outpatients and acute admissions to the emergency department. The former groups were excluded because they are more unlikely to experience the outcome, the latter because the rapid course of events and acute physiology recorded during acute admissions would necessitate a different experimental setup.

## Interpretation

Although ranking the features that drive the predicted risks up or down is useful as a sanity check of the signals picked up by the model, these estimates are not causal. Indeed, the

attribution of each token is determined by its context therefore the same token could be associated with both elevated and diminished risk depending on the other tokens with which it co-occurs. For example, a lab value outside the physiological range (e.g., low creatinine), which would normally drive the risk up, may not affect the predicted risk when co-occurring diagnosis codes counter its contribution (e.g., pregnancy codes).

This is more evident for the free text, where the semantics of a word is always dependent on its context. Overall, the feature attribution does not contain any counterintuitive explanation and the few seemingly questionable interpretations have plausible explanations. For example, we find low eosinophil counts among the lab values associated with elevated risk of clinical deterioration. Clinically, this seems counterintuitive since low values of eosinophils represent the standard and high eosinophil counts are indicative of infections (especially parasitic) and allergic disorders. Eosinophils, however, are part of standard panels for blood differential counts (a count for the different types of white blood cells) and, as such, the very presence of (any) eosinophil count probably reflects a clinical suspicion of infection which may lead to additional analysis to investigate the infectious agent. Interpretation of tokens from the clinical notes also provides some examples that at first seem odd but probably do have some contextual bearing. For example, the tokens *wife* (hustru) and *relatives* (pårørende) are strongly associated with clinical deterioration. This is likely because doctors document in the patient file when relatives have been informed or consulted, and this may well be on poor prognoses or even no-resuscitation orders in which case clinical deterioration is almost certain to ensue.

Numerical features like the number of previous admissions should also be contextualized. While higher age is indeed associated with clinical deterioration, having a lot of admissions prior to the one of the assessments does not. This seemingly paradoxical result can probably be explained by the ability of the model to integrate data from different EHR domains to recognize patients with chronic conditions who will have more frequent hospital visits but perhaps be less likely to suddenly fall critically ill.

## Strengths

This study has some important strengths. First, it is one of the largest of its kind, with a total of 852,620 patients and 2,241,849 admissions taking place over 6 years. Second, we provided a dynamic risk assessment tool that can be used as clinical decision support, showing how the patient's status changes over the course of an admission for different prediction windows and at the same time, providing feature interpretation. This supersedes early warning scores, which are limited to a smaller number of features and do not take sequential information into account but just use a snapshot of the patient's current status. The clinical impact of this model goes beyond its application as decision support tool, as it can potentially be used for prognostic enrichment in clinical trials (enrolling patients depending on their risk of experiencing clinical deterioration) and for resource management to estimate the load of people leaving a hospital department (either because they are transferred to the ICU or die). The real-time implementation of the model in an EHR system is of course dependent on regulatory approvals and on the nature of the EHR. We assume that clinical notes and laboratory values are registered consistently throughout the admission and therefore all the data considered valid at the time of prediction can be used. Third, thanks to the model's architecture, adding new data sources is relatively easy. New features can be added and removed from the model, adapting the tool to the available resources. Healthcare staff have a very busy daily schedule, and it is virtually impossible for them to synthesize hundreds or even thousands of data features regularly. We have shown that the model performs best when as much data as possible is included, likely other data modalities could further improve, for example raw image data, or socioeconomic

data. Another advantage of the architecture of this model is that while a classic EWS depends on complete data to be calculated, we, in this setup, handle implicitly missing data by the entity embedding design, thus allowing assessments also at the doorstep. Finally, unlike existing models, ours does not focus on a specific patient group or medical specialty, but instead can be applied broadly to all patients.

## Limitations

Like any study this has limitations. First, although we tried to define the outcome robustly, there are some pitfalls to consider for ICU transfer and in-hospital mortality. Patients in very severe conditions may still be discharged by the hospital if the latter is not able to provide any support to the patient. These patients will probably experience clinical deterioration within the prediction window but not within their hospital stay, hence they are labelled as negative but potentially still flagged by the model as high-risk patients; this could inflate the number of false positive patients detected by the model. A related limitation is the lack of data on do-not orders; such patients are likely very ill but may not be moved to the ICU and so will introduce noise in the data that complicates training and perhaps hamper performance to some extent. Second, unplanned ICU transfer was captured only for the patients who were admitted in one of the ICUs in our catchment area. It cannot be excluded that some patients admitted to a general department of one of the hospitals in our catchment area are transferred to ICUs of other Danish regions than included here, even if it is uncommon. Third, model performance may improve with additional data such as genetic data, vital signs and other biomarkers. Thanks to the ensemble nature, adding such data in settings where they are available is simple. Fourth, the model was trained on data from the Danish healthcare system and the model (due to the entity embeddings) would likely need to be trained anew if deployed in other geographical or healthcare systems. Finally, vital signs were not used as structured input to the model. Although enabling their full use would have been easier had they been available in tabular form, the model did have access to some information on vital signs insofar as they are recorded in clinical notes, which is especially likely when deviating.

## Conclusion

Combining entity embeddings and recurrent neural networks, we built a highly performant model that every 6 hours during any admission flags patients at higher risk of developing clinical deterioration in the 14 days after the assessment. The model was developed and evaluated using training and validation data, in addition to holdout EHR test data not used during development. A proper prospective evaluation would be needed to establish whether its deployment will produce real-world benefits to patients on hard endpoints. Once clinical utility has been established by trials, the model could both help intervene earlier in patients likely to deteriorate and enrich other clinical trials seeking to identify such early interventions.

## Supporting information

**S1 Fig. Time coverage of the different datasets.**
(TIF)

**S2 Fig. Hyperparameter search results for the model trained only on the DNPR data.**
(TIF)

**S3 Fig. Hyperparameter search results for the model trained only on the biochemical data.**
(TIF)

**S4 Fig. Hyperparameter search results for the model trained only on the clinical notes data.**
(TIF)

**S5 Fig. Calibration curves of the best models trained on all the data for all prediction windows (1, 2, 7, 14 days) and assessment rates (6, 12, 24 hours).**
(TIF)

**S6 Fig. Stratified performances of the best model on AUROC and AUPRC for different age groups.**
(TIF)

**S7 Fig. Stratified performances of the best model on AUROC and AUPRC for different sex.**
(TIF)

**S8 Fig. Data split for model development and evaluation.** Patients are first assigned randomly to either development set or holdout test set (80 and 20%). The development set is used for hyperparameter search (80% assigned for training and 20% for internal validation); the holdout test is used for the assessment of models' performances.
(TIF)

**S1 Table. Hyperparameter selection.** We tested different combinations of the following parameters: bidirectional layer, units, layer, padding size for each data source, dropout, regularization, ICD level characters cut-off, biochemical percentile inclusion, top number of biochemical included, biochemical value.
(PDF)

**S2 Table. Additional metrics for the 12 models at different prediction intervals [1, 2, 7 and 14 days] and different frequencies of assessment [6, 12 and 24 hours].** The metrics are selected for different risk thresholds [1%, 5%, 10%, 20% and 50%]. Statistics are provided as median and 95% confidence interval: (median, 2.5% percentile, 97.5% percentile).
(PDF)

**S3 Table. Number of patients, admissions, and positive cases at different times of prediction.** Given the same prediction window, the statistics for the models at lower frequency of assessment (12 and 24 hours) would be the same as the model presented in this table (e.g., the model with a frequency of assessment of 12 hours and a prediction window of 14 days has the same statistics of the model having the same prediction window and a frequency of assessment of 6 hours).
(PDF)

**S4 Table. TRIPOD statement.**
(PDF)

## Author Contributions

**Conceptualization:** Davide Placido, Hans-Christian Thorsen-Meyer, Benjamin Skov Kaas-Hansen, Søren Brunak.

**Data curation:** Davide Placido.

**Formal analysis:** Davide Placido.

**Funding acquisition:** Søren Brunak.

**Investigation:** Davide Placido, Søren Brunak.

**Methodology:** Davide Placido, Hans-Christian Thorsen-Meyer, Benjamin Skov Kaas-Hansen, Roc Reguant.

**Project administration:** Søren Brunak.

**Resources:** Søren Brunak.

**Software:** Davide Placido, Hans-Christian Thorsen-Meyer, Benjamin Skov Kaas-Hansen, Roc Reguant.

**Supervision:** Hans-Christian Thorsen-Meyer, Søren Brunak.

**Validation:** Davide Placido.

**Visualization:** Davide Placido, Benjamin Skov Kaas-Hansen.

**Writing – original draft:** Davide Placido, Benjamin Skov Kaas-Hansen.

**Writing – review & editing:** Davide Placido, Hans-Christian Thorsen-Meyer, Benjamin Skov Kaas-Hansen, Roc Reguant, Søren Brunak.

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
