## [Decision Letter · Decision Letter 0]

2 Dec 2022

PDIG-D-22-00253

Development and validation of a dynamic prediction model for unplanned ICU admission and mortality in hospitalized patients

PLOS Digital Health

Dear Dr. Brunak,

Thank you for submitting your manuscript to PLOS Digital Health. After careful consideration, we feel that it has merit but does not fully meet PLOS Digital Health's publication criteria as it currently stands. Therefore, we invite you to submit a revised version of the manuscript that addresses the points raised during the review process.

EDITOR: Please insert comments here and delete this placeholder text when finished. Be sure to:

* Indicate which changes you require for acceptance versus which changes you recommend

* Address any conflicts between the reviews so that it's clear which advice the authors should follow

* Provide specific feedback from your evaluation of the manuscript

Please submit your revised manuscript within 60 days Jan 31 2023 11:59PM. If you will need more time than this to complete your revisions, please reply to this message or contact the journal office at digitalhealth@plos.org. Please include the following items when submitting your revised manuscript:

We look forward to receiving your revised manuscript.

Kind regards,

Martin G Frasch

Section Editor

PLOS Digital Health

Journal Requirements:

1. Please send a completed 'Competing Interests' statement, including any COIs declared by your co-authors. If you have no competing interests to declare, please state "The authors have declared that no competing interests exist". Otherwise please declare all competing interests beginning with the statement "I have read the journal's policy and the authors of this manuscript have the following competing interests:"

a. Please clarify all sources of funding (financial or material support) for your study. List the grants (with grant number) or organizations (with url) that supported your study, including funding received from your institution. 

b. State the initials, alongside each funding source, of each author to receive each grant.

c. State what role the funders took in the study. If the funders had no role in your study, please state: “The funders had no role in study design, data collection and analysis, decision to publish, or preparation of the manuscript.”

d. If any authors received a salary from any of your funders, please state which authors and which funders.

3. We ask that a manuscript source file is provided at Revision. Please upload your manuscript file as a .doc, .docx, .rtf or .tex.

4. We have noticed that you have uploaded Supporting Information files, but you have not included a list of legends. Please add a full list of legends for your Supporting Information files after the references list. 

5. Since your data is not available for proprietary reasons, please explain via email why the data is not available. Please also include the contact information for the third party organization that should be contacted should other researchers want to request access to this data and please include the full citation of where the data can be found. We also request that you verify with us via email that any researcher will be able to obtain the data set in the same manner that the you have obtained it. If you feel you are unwilling or unable to adhere to this policy, please explain your reasons by return email and your exemption request will be escalated to the editor for approval. Your exemption request will be handled independently and will not hold up the peer review process, but will need to be resolved should your manuscript be accepted for publication. One of the Editorial team will be in touch if they require more information.

Additional Editor Comments (if provided):

Reviewers' comments:

Reviewer's Responses to Questions

**Comments to the Author**

1. Does this manuscript meet PLOS Digital Health’s publication criteria? Is the manuscript technically sound, and do the data support the conclusions? The manuscript must describe methodologically and ethically rigorous research with conclusions that are appropriately drawn based on the data presented.

Reviewer #1: Partly

Reviewer #2: Yes

2. Has the statistical analysis been performed appropriately and rigorously?

Reviewer #1: Yes

Reviewer #2: Yes

3. Have the authors made all data underlying the findings in their manuscript fully available (please refer to the Data Availability Statement at the start of the manuscript PDF file)?

Reviewer #1: No

Reviewer #2: Yes

4. Is the manuscript presented in an intelligible fashion and written in standard English?

Reviewer #1: Yes

Reviewer #2: Yes

5. Review Comments to the Author

Reviewer #1: Overall summary: This retrospective cohort study from Placido and colleagues aimed to develop a deep learning early warning system for clinical deterioration on the hospital wards across multiple hospitals in Denmark. The authors demonstrated high discrimination for a composite outcome of ICU admission or death.

Major Strengths

1. The authors used an extremely large cohort across multiple hospitals, which will help with generalizability and replication.

2. Isotonic recalibration is a nice approach to ensuring that the model does well across multiple performance measures (i.e., beyond just AUROC).

Major Limitations

1. It’s not entirely clear to me where the novelty lies in this study. The Rajkomar study (ref # 14) has in some ways set the bar for hospital outcome prediction using deep learning from the EHR, and while this study purports to be more of an early warning system, designing predictions for a 14 day window is not much of an early warning. I think the study rationale/objectives should be more clearly articulated and more strongly argued.

2. The authors’ methods are nice from the modeling perspective, but it’s less clear how clinically operationalizable they will be. In particular, tokenizing clinical notes is fraught with uncertainty in terms of implementation and use. At what point in a real-time system would a note be “scanned” or “re-scanned”? What happens to notes written by house officers or APPs which are awaiting attending physician cosignature? What happens to notes which are revised, amended, or deleted?

3. How were missing data handled in this approach?

4. Lines 88-97 don’t make sense to me. The authors argue that vital signs might be unreliable as EWS inputs due to limited recording, but then state that they used EHR data and clinical notes – which are either (a) exactly those same vital signs just discounted or (b) even more sparse than ward vital signs (e.g., once-daily notes). Can the authors reconcile?

5. I note that TRIPOD is mentioned in the Methods, but I do not see a relevant checklist attached in the Supplement.

Minor comments:

1. Figure 1 is not referenced in the manuscript

2. Line 322 seems incomplete: “performances of h is not…” (what is “h”?)

Reviewer #2: Development and validation of a dynamic prediction model for unplanned ICU admission and mortality in hospitalized patients

Placido et al. presented a manuscript in which they developed a dynamic prediction model with a large dataset including almost all hospitalized patients from 12 hospitals in Denmark. The authors used deep learning techniques to create the best model for the prediction of clinical deterioration. In general, the manuscript is well written and the study seems to be well performed. However, there are several concerns that need to be addressed:

Our main concern is the clinical impact of this study for the patient and/or health care provider. While reading this manuscript, the following questions arose: What is the advantage for patients/clinical care when using this model? How should the caregiver use this model in practice? What does this model add to the existing deep learning and conventional ‘epidemiological’ prediction models and literature? The manuscript mainly focusses on explaining their methods/the deep learning method. We recommend to add the clinical perspective, importance and integration of the model to the manuscript.

Our second concern relates to the data handling which is not reported. Did someone perform data cleaning on this big dataset? How were outliers handled? The authors mention in their introduction that caregivers make mistakes in manually collecting data. However, part of your data were likely to be collected or reported manually by caregivers. How much data were missing? And how did the authors handle missing data? Please add this to the manuscript.

The other comments are listed below in a consecutive order:

Title:

- The authors only performed internal validation using their own dataset. Therefore, the word ‘validation’ is too broad and not appropriate in this setting. We suggest to change it to ‘internal validation’ or remove it completely. Also, the authors can provide more details on the target population. Please consider adding the country of the study population.

Abstract:

- The abstract is has been disproportionally arranged. Information about the statistical/deep learning methods used is lacking, while other parts of the abstract (introduction) are relatively long. Please rearrange the abstract and add more details on the methods and the time points used for dynamic prediction.

- The final sentence of the manuscript (line 44-46) does not match the main scope and overall conclusion of the manuscript. Please rewrite this.

- Please spell out the abbreviations used in the abstract when reported for the first time.

Introduction:

- The introduction should start with a broader introduction of the topic. What is the importance of prediction? What are the limitations of currently used risk scores and existing deep learning models? What could the model add to clinical decision-making, i.e. what are the advantages for the patient and heath care professionals? Please explain the rationale for developing the dynamic prediction model using deep-learning and add more information on what this model could add.

- The only prediction model mentioned is the EWS. However, there are many more prediction models which are often developed in a department or for a specific patient group. The authors could first write something about prediction in general, and later introduce the EWS score, as an example.

- The modified early warning score (MEWS) is more often used than the EWS nowadays. Please explain why you compare to the EWS.

- Please state the research question and objectives more specifically at the end of the introduction. 

- Please mention that this is a model development study with internal validation. Moreover, a hypothesis is missing. Please add this.

- Figure 1: this figure needs clarification. The 2 different examples are now difficult to distinguish. Separation of the 2 examples could be clarifying. Furthermore, the numbers/hours/days of the time points, the prediction window (particularly for ICU transfer), and outcomes used should be clarified in the figure.

Methods:

Major concerns

- In general, the deep-learning analyses seem to be performed appropriately. It would be valuable to incorporate more metrics in the supplemental material, such as the precision and recall loss. Please comment on this suggestion.

- Please define the data used for developing the model more extensively. More information on which medical diseases, biochemical measures, and clinical notes are used in the dynamic model development is essential.

- There is no missing data section in the methods. Please elaborate on the number of missing data, how missings were handled, and which choices or assumptions have been made. For example:

o Not all biochemical data were available every day and at the specific time points, how did the authors deal with this?

o Which clinical notes were used? Daily ones?

o How much data were missing at the specific time points? And how many patients were included at different time points?

o What did the authors do with patients with a non-ICU policy? These patients are too ill, too old, or have to many comorbidities to be admitted to the ICU and could be more at risk for in-hospital mortality and theoretically ICU admission, whereas they are not allowed to be admitted to the ICU. Did the authors take this into account? Please consider a sensitivity analysis.

- Table 1, there are a few comments on this table:

o Please move this table to the results section, as it describes the characteristics of the patients.

o Please provide more information about patient characteristics, particularly clinical data. What are the comorbidities/chronic illnesses of the patients? Which department are they from? Why are they admitted? This is a lot of information that the authors can also decide to add to the supplement.

o Please clarify what the numbers in between the brackets mean.

o What does the 'Number of previous admissions' mean? How is it possible that 3 [1-7] people died in the hospital? Please clarify this.

o Has any data cleaning been performed? Did the authors check for outliers?

- What did the authors do with patients who died after, or in between 2 moments of, the prediction window? Are they not included? Please specify the numbers of patients that were included in each model (each frequency of assessment and prediction window) in the results.

- Did the data include patients from ALL wards, except the ICU and Emergency Department? Can the authors please specify or add a list with the wards of which patients are included? For example, did the authors also include patients from the ophthalmology and ear-nose-throat departments?

- Please add information on who assessed or reported the outcome measures? Were those researchers blinded?

Minor concerns:

- Line 113: please specify the definition of ‘direct ICU admissions’. Are those patients admitted within hours? Or directly?

- Line 120: this is unclear. It seems that those patients have already been discharged and are then admitted again. Please reformulate or clarify this.

- Line 121: please adjust the title of this paragraph. A suggestion could be ‘model development’.

- Figure 2: please enlarge this figure, the figure is difficult to read.

- Line 189-194: maybe the authors can add a figure to the supplements to visualize this process.

Results:

- Please add data on how many patients were included per prediction window and per time point? This is interesting because a lot of patients are in the hospital for less than 14 days. What happened if a patient died or was discharged outside the prediction model? This is also stated in the TRIPOD guidelines, 14a: ‘Specify the number of participants and outcome events in each analysis.’

- Calibration is a very important component of the predictive performance of a prediction model. Figure S5, the calibration plot of the best performing model, should be replaced to the main paper. In addition, please visualize all other calibration plots in the supplement. Furthermore, please add more textual information about the calibration of the models and their interpretation, as this is now underreported.

- Please explain more why the 6-hour/14-day model is the best model. This is not clear now as it has not the highest AUROC or best calibration. Only the AUPRC, as a measure of discrimination, is highest.

- Figure 4DEF: these figures are very interesting and important, and should be highlighted more. Please adjust these, since there are unreadable.

Discussion:

Major concerns

- In the discussion, the authors should focus more on the clinical interpretation. Why is this model better than already existing/developed models and how can we apply the model in daily patient care? How do the authors envision this?

- The paragraph 'interpretation' should be added to the limitations paragraph, as the content of this paragraph contains limitations instead of interpretations.

- Limitations:

o The authors describe that there are many errors in the manual collection of data, but there are also errors in data that has been pulled directly from the system. How did they deal with this?

o Vital signs have been left out, but are very important for mapping a patient, assessing clinical deterioration/improvement and estimating ICU requirement. In addition, they are likely to be measured at all departments. Regardless of whether vital signs are included in the model, it is important to still measure vital signs as clinical parameter. Please add this to the limitations.

o Add a section regarding missing data.

- Strengths: this model differs from other models, because it does not select a specific patient group because with a certain disease or at a specific department. We assume that all patients admitted to all departments are included. Existing models often aim to predict at a certain group of patients or a certain department. This is a strength of the model that could be emphasized.

- Please discuss the potential clinical use of the model and implications for future research in a new paragraph. Consider external validation.

- The conclusion should apply more directly to the study, presenting the best model, results, and implication of this study. Please incorporate this.

Minor concerns:

- Line 316: number 0.28 is incorrectly rounded, please change it to 0.29

- Line 322: there is one word or there are a few words missing in this sentence. What does h mean? Please rewrite.

6. PLOS authors have the option to publish the peer review history of their article (what does this mean?). If published, this will include your full peer review and any attached files.

**Do you want your identity to be public for this peer review?** For information about this choice, including consent withdrawal, please see our Privacy Policy.

Reviewer #1: No

Reviewer #2: Yes: Iwan van der Horst

---

## [Decision Letter · Decision Letter 1]

24 Apr 2023

Development of a dynamic prediction model for unplanned ICU admission and mortality in hospitalized patients

PDIG-D-22-00253R1

Dear Professor Brunak,

We are pleased to inform you that your manuscript 'Development of a dynamic prediction model for unplanned ICU admission and mortality in hospitalized patients' has been provisionally accepted for publication in PLOS Digital Health.

Best regards,

Jessica Keim-Malpass

Academic Editor

PLOS Digital Health

Reviewer Comments (if any, and for reference):

Reviewer's Responses to Questions

**Comments to the Author**

1. If the authors have adequately addressed your comments raised in a previous round of review and you feel that this manuscript is now acceptable for publication, you may indicate that here to bypass the “Comments to the Author” section, enter your conflict of interest statement in the “Confidential to Editor” section, and submit your "Accept" recommendation.

Reviewer #1: All comments have been addressed

2. Does this manuscript meet PLOS Digital Health’s publication criteria? Is the manuscript technically sound, and do the data support the conclusions? The manuscript must describe methodologically and ethically rigorous research with conclusions that are appropriately drawn based on the data presented.

Reviewer #1: Yes

3. Has the statistical analysis been performed appropriately and rigorously?

Reviewer #1: Yes

4. Have the authors made all data underlying the findings in their manuscript fully available (please refer to the Data Availability Statement at the start of the manuscript PDF file)?

Reviewer #1: No

5. Is the manuscript presented in an intelligible fashion and written in standard English?

Reviewer #1: Yes

6. Review Comments to the Author

Reviewer #1: Thank you for the thorough responses to my comments and those of the other reviewers. The manuscript is markedly stronger as a result. I think this work represents a nice proof of concept and opens some important conversations regarding how we might operationalize some of these approaches in the real world.

7. PLOS authors have the option to publish the peer review history of their article (what does this mean?). If published, this will include your full peer review and any attached files.

**Do you want your identity to be public for this peer review?** For information about this choice, including consent withdrawal, please see our Privacy Policy.

Reviewer #1: **Yes: **Patrick G Lyons
